# Doxycycline Prevents Preclinical Atherosclerosis, Pancreatic Islet Loss and Improves Insulin Secretion after Glycemic Stimulation: Preclinical Study in Individuals with a High-Fat Diet

**DOI:** 10.3390/biomedicines11030717

**Published:** 2023-02-27

**Authors:** Alejandrina Rodriguez-Hernandez, Marina Delgado-Machuca, Rodolfo Guardado-Mendoza, Martha A. Mendoza-Hernandez, Valery Melnikov, Osiris G. Delgado-Enciso, Daniel Tiburcio-Jimenez, Gabriel Ceja-Espiritu, Gustavo A. Hernandez-Fuentes, Armando Gamboa-Dominguez, Jose Guzman-Esquivel, Margarita L. Martinez-Fierro, Iram P. Rodriguez-Sanchez, Ivan Delgado-Enciso

**Affiliations:** 1School of Medicine, Universidad de Colima, Colima 28040, Mexico; 2Cancerology State Institute, Colima State Health Services, Colima 28085, Mexico; 3Department of Medicine and Nutrition, Universidad de Guanajuato, Campus León, León 37320, Mexico; 4Clinical Epidemiology Research Unit, Mexican Institute of Social Security, Instituto Mexicano del Seguro Social, Villa de Álvarez 28984, Mexico; 5Belisario Domínguez Sección XVI, Pathology Department, Instituto Nacional de Ciencias Médicas y Nutrición Salvador Zubirán, Mexico City 14080, Mexico; 6Molecular Medicine Laboratory, Unidad de Medicina Humana y Ciencias de la Salud, Universidad Autónoma de Zacatecas, Zacatecas 98160, Mexico; 7Molecular and Structural Physiology Laboratory, School of Biological Sciences, Universidad Autónoma de Nuevo León, San Nicolás de los Garza 66455, Mexico

**Keywords:** doxycycline, diabetes, treatment, pancreas, atherosclerosis

## Abstract

Doxycycline (Doxy) is an antibiotic, which has exhibited anti-inflammatory activity and glucose metabolism improvement. The present study was proposed to evaluate its effects on glucose metabolism and other associated processes, such as lipemia and adipogenesis, as well as, to evaluate its effects on the liver, pancreas, and aorta in subjects fed with an occidental high-fat diet (HFD). The trial followed three groups of BALB/c mice for 6 months: (1) Standard diet (SD); (2) HFD-placebo (saline solution); and (3) HFD-Doxy (10 mg/kg/day). Intrahepatic fat accumulation (steatohepatosis) and the epididymal fat pad, as well as the hepatic inflammatory infiltrate and ALT serum levels were higher in both groups with the HFD (with/without doxycycline) in comparison with the SD group. The thickness of the aorta (preclinic atherosclerosis) was significantly elevated in the HFD group with respect to the HFD + Doxy and SD group, these two being similar groups to each other. The HFD-Doxy group had pancreatic morphological parameters very similar to those of the SD group; on the contrary, the HFD group reduced the number of pancreatic islets and the number of β cells per mm^2^, in addition to losing large islets. The index of β cell function (∆Insulin0–30/∆Glucose0–30 ratio) was significantly higher in the HFD + Doxy group, compared to the rest of the groups.

## 1. Introduction

Diabetes mellitus is becoming one of the main threats to human health in the 21st century [1]. This disease has been described as a metabolic disorder with multiple etiologies, characterized by alterations in carbohydrate, fat, and protein metabolism due to insulin production anomalies. 

There are two principal types of diabetes: type 1 diabetes mellitus (DM1), which is due to the autoimmune destruction of beta pancreatic cells, and type 2 diabetes mellitus (DM2) that implies anomalies in secretion and insulin activity. Ideal control of DM1 and advanced states of DM2, require exogenous insulin administration. However, the search for medications to prevent or protect the pancreatic tissue against this disorder is still an ongoing process. It has been postulated that some immunotherapies could delay the deterioration of beta cell pancreatic activity. Particularly, the use of teplizumab, a monoclonal antibody against CD3, could significantly delay (two years interval) the clinical onset of DM1 in selected patients [2].

Doxycycline is a semi-synthetic antibiotic, a member of the tetracycline family, with a broad bacteriostatic spectrum due to its inhibition of protein synthesis by avoiding the binding of bacterial ribosomic units [3]. Periodontitis, an infection that often coexists with severe diabetes mellitus, is commonly treated with doxycycline. Previous studies suggest that doxycycline improves metabolic control in diabetes mellitus [4]. In addition, there have been isolated case reports of youth without diabetes treated with doxycycline who subsequently experienced hypoglycemia [5]. Another study reports that antibiotics could prevent the onset of type 1 diabetes mellitus in mice models prone to DM1; this effect has been attributed to the drug’s bacteriostatic properties, which decrease the stimulation of the immune system [6]. These results have been replicated previously in studies that use doxycycline in a diabetic non-obese mouse model [7].

Other studies in male db/db mice show that doxycycline administration supplemented in drinkable water for 10 weeks improves glucose and insulin tolerance. It also produces an increase in the number of pancreatic islets and beta cell percentage [8]. The db/db mice model (model with leptin gene mutation) is the most used model for DM2 [9]. However, the genetic nature of this model prevents the comparison with healthy individuals; therefore, it was not possible to establish if doxycycline administration can maintain pancreatic morphology and glucose metabolism in similar parameters in healthy individuals. In human beings, a recent clinical trial in which 100 mg/kg/day of doxycycline was administered for 10 weeks, showed reduced inflammation and improved muscular sensitivity to insulin in people with DM2 [7]. Diabetes confers about a two-fold excess risk of coronary heart disease, stroke, and death due to other vascular causes [10]. It has been demonstrated that doxycycline administration could reduce the formation of atherosclerosis or its pathophysiologic process in different mice models generated through mechanic endothelial damage, bacterial infection, or a high-fat diet. In accordance with this information, it has been observed that doxycycline benefits in cardiovascular disease and other related chronic inflammation disorders [11,12,13]. However, the anti-atherosclerotic effect of doxycycline has not been demonstrated in mice fed with a high-fat diet in a non-transgenic mice model [14,15,16], which would better represent the pathophysiology of the disease in humans. Therefore, it is necessary to improve the knowledge that facilitates finding new therapeutic approaches in atherosclerosis prevention and treatment.

The present work was conducted to assess if the chronic administration of doxycycline in non-transgenic mice with an occidental high-fat diet could prevent simultaneously atherosclerosis, steatosis, and hepatic inflammation [13,17], as well as pancreatic morpho-functional alterations that deregulate glucose metabolism. In addition, this work also sought to evaluate its effect on serum lipids, adipogenesis, and body weight.

## 2. Materials and Methods

### 2.1. Reactive and Animals Treatment

Doxycycline was purchased from Sigma Aldrich (D9891, Saint Louis, MO, USA). A standard diet was used (2018S Tekland Global 18% Protein Rodent Diet, Harlan^®^, Madison, WI, USA) containing 18.6% protein, 46.9% carbohydrate, and 13.2% fat. The high-fat diet used (TD.02028 Atherogenic Rodent Diet, Harlan^®^, Madison, WI, USA) contains 17.3% protein, 46.9% carbohydrate, and 21.2% fat. 

Forty-two BALB/C male mice (Harlan^®^, Mexico City, Mexico) aged between 4 and 6 weeks of life and with an initial weight between 22 and 25 grammes were used in this study. Two groups (thirteen mice each) were fed with an occidental high-fat diet (HFD, Atherogenic Rodent Diet, Harlan^®^, Madison, WI, USA). One of them was administered doxycycline to conform to the experimental group (HFD + Doxy group), and the other one was left untreated (HFD group); this last one served as a reference of the pathological alterations produced by the high-fat diet. A third group of sixteen mice was fed with the standard diet (2018S Tekland Global 18% Protein Rodent Diet, Harlan^®^, Madison, WI, USA); this group was assigned as a reference of the healthy mice (Standard diet, SD, group).

The mice were maintained in cages in groups with a maximum of five mice under standard light and temperature-controlled conditions with food and water ad libitum. The drug was administered in dissolved water, as previously described, with a calculated consumption equal to a dose of 10 mg/kg per day. The daily ingestion of water and the mice weights were measured every 15 days, with the purpose of re-calculating the drug concentrations in the drinking water to maintain the established dose. The supplemented water was changed each 48 h. The diet and treatment were administered for 6 months, at the end of which a glucose tolerance curve was made and the mice were sacrificed by decapitation. Blood samples were collected for biochemical analyses, and the pancreas, liver, and aorta were extracted for histological analyses.

In this study, a model of preclinical atherosclerosis and non-alcoholic steatohepatitis was employed with genotype native mice (BALB/c). The mice were subjected to a high-fat diet for a period of 6 months, inducing metabolic dysregulation [13]. This mouse model is ideal for evaluating this aspect as it intervenes in a slow development of atherosclerosis and non-alcoholic steatohepatitis in the initial stages [18]. This and other similar models of non-transgenic mice do not develop significant increases in body weight, but do develop morpho-histological and metabolic changes in the early stages of chronic diseases associated with high-fat diets [19,20,21,22].

All animal procedures were handled in accordance with institutional guidelines and the Official Mexican Standard that regulates the use of laboratory animals (NOM-062-ZOO-1999), in addition to the guidelines of the American Veterinary Medical Association 2020 for the slaughter of animals. This protocol was approved by the ethical and investigation committee of the School of Medicine in the Colima University (Colima, Colima, México).

### 2.2. Histopathological Analysis

Pancreatic tissue was fixed with 10% formaldehyde, dehydrated in ethanol, embedded in paraffin to be sectioned (5 µm thickness), dewaxed at 100 °C for 20 min, and transferred to xylene, ethanol–xylene, and absolute ethanol solutions, and finally washed in distilled water for immunohistochemical staining. Pancreatic immunohistochemical staining was done using antibodies against insulin (clone HB125), as previously reported [23]. Additionally, the anti-PD1 antibody (Clone IHC001) (programmed cell death protein-1 antibody) and anti-Ki67 antibody (Clone K-2) (100 mL monoclonal mice antibodies at 1:200 dilution; Biogenex, San Ramon, CA, USA) were used, according to the previous reported indications [23]. Each immunohistochemical process was done individually (not combined); as a result, it is not feasible to identify if the positive cells Ki67 or PD-1 were positive to insulin too.

The marked pancreatic sections were chosen at random and histomorphometric studies were performed with a digital camera model Axiocam MRC-5 (Zeiss, Göttingen, Germany) connected to a brightfield optical microscope model AxioPlan 2 M (Zeiss^®^, Göttingen, Germany) with a motorized stage and an A-plan 10x, 20x and 40x objective (total magnification of X100, X200, and X400). Using MosaiX and Autofocus modules, images of all sample surfaces were scanned, and islet areas were measured using contour spline. All images were obtained under the same lighting conditions and exposure times in the imaging program AxioVs 40 V.4.7.0.0 (Carl Zeiss Imaging Solutions GmbH, imaging program 2006-200, Munich, Germany).

Three sections of pancreatic tissue per mice were analyzed and each one of three areas (anterior, medial, and posterior) were measured. The number of islets per mm^2^ was determined as previously described [24]. For the islet size distribution analyses, the islets smaller than 10.000 µm^2^ were designated as “small”, islets larger than 10.000 µm^2^ but smaller than 25.000 µm^2^ were designated as “medium”, and the ones larger than 25,000 µm^2^ were designated as “big” [25].

The total number of beta cells per mm^2^ was obtained by counting the nuclei surrounded by cytoplasmic insulin immunostaining. The insulin-positive cells were measured based on their area positive of insulin divided by their nuclei number. Additionally, the area covered by the nuclei in beta cells, was determined to calculate the nuclei and cytoplasmatic area. Inside each islet, the area covered with immunostaining insulin-positive cells, Ki67 and PD-1 were measured, in which the percentage of islet area covered for beta cells or for cells with proliferative and apoptosis markers.

Liver and aorta processing was done as previously reported [17]. Hepatic inflammation and steatohepatosis were evaluated, checking the percentage of hepatic tissue with inflammation and fat accumulation. Both steatohepatosis and hepatic inflammation were classified as grade 0 (absent), grade 1 (until 33%), grade 2 (between 33% and 66%), and grade 3 (more than 66%), and also were identified as mild, moderate, and severe [13,17,25,26]. The abdominal aorta was analyzed in three portions (proximal, medial, and distal). A quantitative evaluation of the atherosclerosis was made by measuring the thickness of the medium intimacy (from the inner endothelial layer to the outer of the medial layer). A systematic randomized sampling was used to select eight sections equidistant by section, regardless of whether atherosclerotic lesions were present or absent [13,17].

### 2.3. The Body and Liver Weight, and Epididymal Fat Pads

Body and liver weight were obtained to calculate the liver–body weight ratio, which has been reported to increase in animal models with hypercaloric diets [27]. Furthermore, the fat pad of the epididymis was determined, as it provides a good parameter to study alterations in rodent fat accumulation (the adipogenic effect, which occurs when fat accumulation increases) [28]. 

### 2.4. Glucose Tolerance Curve

A glucose tolerance curve was measured intraperitoneally at 6 months. Animals were kept fasting overnight, and glucose was injected IP (2 mg/kg of body weight). Glucose and insulin were measured in blood samples at 0, 30, 60, 90, and 120 min after administration. A venous blood sample was collected from each mice tail. The area under the insulin curve (iAUC) and the incremental glucose area under the curve (gAUC) were calculated using the trapezoidal method. Early response to insulin (∆Insulin0-30/∆Glucose0-30 ratio), also known as the insulinogenic index, was calculated as previously described [29]

### 2.5. Biochemical Analyses

Before the slaughter, blood samples were collected from the mice after 6 h of fasting to measure serum lipids (triglycerides, total cholesterol) and liver enzymes (ALT, AST), using an automatic biochemical analyzer (Cobas c111, Roche ^®^, Mexico City, Mexico). A commercial ELISA kit was used for insulin detection (EZRMI-13K, Millipore, Darmstadt, Germany).

### 2.6. Statistical Analyses

Data was collected and analyzed using the SPSS statistical software (Version 20, IBM, Armonk, NY, USA: IBM Corp., Chicago, IL, USA). The Komolgorov–Smirnov test was realized to evaluate the normality distributions of the data. Unidirectional analysis of variance (ANOVA) was used to compare data with a normal distribution, using Dunnett’s post hoc test. Post hoc Kruskal–Wallis and Mann–Whitney U tests were used to compare data with a non-normal distribution.

## 3. Results

### 3.1. Biochemical and Histopathological Analysis

Table 1 shows values of various parameters at the end of the study and a statistical comparison between mice fed the standard diet (SD), high-fat diet (HFD), or HFD plus doxycycline (HFD + Doxy). Table 2 shows a “post hoc” analysis of the parameters that were significantly different shown in Table 1 (fat accumulation, inflammation, and preclinical atherosclerosis), making comparisons such as SD vs. HFD, SD vs. HFD + Doxy, or HFD vs. HFD + Doxy.

According to the results, body weight and serum lipid profile were similar between groups. Doxycycline administration in the high-fat diet groups did not produce significant changes in feed intake (Table 1). The accumulation of liver fat (steatosis) and epididymal fat pad was higher in the two groups with a high-fat diet (with and without doxycycline), compared to the group fed with a standard diet (see Table 1 and Table 2). The high-fat diet caused an elevation of parameters associated with liver inflammation, such as hepatic inflammatory infiltrate and ALT levels. In the HFD-diet mice, administration of doxycycline reduced steatosis, inflammatory infiltrate, and serum ALT levels, although not significantly. Another variable associated with chronic systemic inflammation, such as thickening of the aorta (preclinical atherosclerosis), was significantly elevated in the HFD group, compared to the HFD + Doxy or SD groups, the latter two being similar groups between them (see Table 1 and Table 2). Given the above, one can conceive that doxycycline is not able to reduce the adipogenic effect, steatosis, hepatic inflammatory infiltrate, or alterations of liver enzymes, caused by a diet rich in fat, but it could prevent preclinical atherosclerosis generated by this diet.

### 3.2. Pancreatic Morphology

Mice fed with the high-fat diet receiving doxycycline had pancreatic parameters similar to those fed with the standard diet (see Table 1 and Table 2). In contrast, the high-fat diet reduced the number of pancreatic islets and the number of β cells per mm^2^, also losing the large islets (Figure 1). Cell size was increased in individuals with a high-fat diet (with and without doxycycline) with respect to those fed with a standard diet, being the only aspect of the pancreatic morphology in which the SD and HFD + Doxy groups differed. The size of the islets and the percentage of area occupied by β cells on each islet were similar in all groups. The HFD group had a significantly higher percentage of proliferating cells in its pancreatic islets (Ki67-positive cells) in comparison to the SD or HFD + Doxy groups, as well as a higher proportion of cells in apoptosis (PC-1-positive cells), but without being statistically significant.

### 3.3. Glucose Metabolism

No differences in fasting serum glucose were observed between groups. Fasting insulin levels were remarkably increased in the HFD + Doxy group, compared to the standard diet group (see Table 1 and Table 2). On the glucose tolerance curve, the HFD + Doxy group had significantly lower blood glucose at 30 min than other groups, including the group fed with the standard diet. In contrast, the high-fat diet group (without doxycycline) had a significantly elevated blood glucose at 30 min compared to the rest of the groups (see Figure 2A). Serum insulin levels were elevated at baseline in the doxycycline group and incremented greatly at 30 min (see Figure 2B), according to its lower blood glucose levels at the moment of the trial. The HFD + Doxy group had a significantly lower gAUC 0–120 min and a significantly higher iAUC/gAUC 0–90 min ratio than the HFD group (without doxycycline) (see Figure 2C,D). The β cell function index (ratio ∆Insulin0–30/∆Glucose0–30) was significantly higher in the HFD + Doxy group compared to the rest of the groups, while the HFD group (without doxycycline) had the lowest values between groups (see Figure 2F). This indicates that high-fat chronic diets reduce the ability of β cells to release insulin into the blood, while doxycycline administration increases it significantly.

## 4. Discussion

Administration of doxycycline prevents the loss of pancreatic islets and β cells in individuals chronically fed with a high-fat diet. Additionally, doxycycline increases the ability of β cells to release insulin after a glycemic stimulus. This indicates a dual effect of doxycycline, possibly acting as a protector of the morphology of the pancreatic islets and as an enhancer of the function of the β cells. 

It should be noticed that the non-transgenic mice model fed with a high-fat diet used in these experiments allowed the evaluation of preclinical stages of diabetes [30] and atherosclerosis [31]. It is important to mention that in the preclinical stages of metabolic disorders, the fasting serum parameters of lipids, glucose, and insulin may not present alterations, so it is necessary to evaluate other more predictive or sensitive tests to establish differences between individuals [18,32,33,34]. In the present work, the parameters of fasting glucose, cholesterol, and triglycerides that were evaluated did not present differences between the groups. However, notable benefits were found in the group treated with doxycycline in the early stages of metabolic disorders and preclinical atherosclerosis.

Further to its antibacterial properties, doxycycline also exhibits anti-inflammatory, anti-apoptotic, and antioxidant properties [35]. Therefore, it has been postulated as a potential treatment of diseases associated with chronic inflammation through the non-selective inhibition of matrix metalloproteinases (MMPs) [36]. Additionally, it has been postulated that treatment with a sub-antimicrobial dose is able to reduce the inflammatory mediator’s serum concentration and thereby lower the risk of onset of cardiovascular diseases [37]. The relationship between atherosclerosis and diabetes has been widely studied, although, atherosclerosis is generally assumed to be a result of processes triggered by diabetes [38]. However, the present study shows that damage to pancreatic cells and atherosclerosis are generated simultaneously in the context of a high-fat diet and that doxycycline is able to prevent both alterations.

Doxycycline is a semi-synthetic antibiotic, a member of the tetracyclines, that has proved to improve the metabolic control of diabetes [4]. Furthermore, there are isolated cases of non-diabetic youth with doxycycline-induced hypoglycemia [5], which is compatible with the increase in serum insulin generated due to doxycycline [6,7,8].

The study by Wang et. al., (2017) in a db/db mice model of leptin deficiency (with a mutation in the gene encoding the leptin receptor), found that doxycycline increases the number and percentage of pancreatic islets of β cells, while decreasing the size of the islets. Unlike the report of Wang et. al., (2017), the results of the present study indicate that the effect of doxycycline on the pancreatic islets and beta cells is due to the protection against “the loss” caused by the high-fat diet, thus, preventing the loss of pancreatic tissue, and not due to an increase in pancreatic cell mass. We can also mention that the size of the islets is not reduced by doxycycline, and also, it prevents the loss of large islets due to the consumption of a diet rich in fat, remembering that large islets are the ones that are preferentially lost in DM2 [39]. In general, the use of doxycycline avoids deleterious histological changes caused by the high-fat diet, maintaining histological parameters at values similar to those of healthy individuals (with a balanced diet). The only histologic aspect of the pancreas that varies between an individual on a balanced diet and an individual on a high-fat diet plus doxycycline is the size of the β cell, which is significantly larger in the doxycycline group. This could be compatible with some degree of beta cell hypertrophy, which would be consistent with higher insulin levels, baseline and during the glucose tolerance curve, in the HFD + Doxy group than in the standard diet group (see Figure 2). This is manifested in a significantly higher index of β cell function (∆Insulin0–30/∆Glucose0–30 ratio) in the HFD + Doxy group, compared to the rest of the groups in this study (see Figure 2). This indicates that chronic diets high in fat reduce the ability of β cells to release insulin into the blood, while the administration of doxycycline increases it significantly. In the animal model used in the present study, doxycycline administration did not reduce body weight, the liver–body weight ratio, nor the liver fat accumulation or serum lipids. Liver inflammation tends to be minor, but not significantly. The former suggests that doxycycline’s effects on atherosclerosis and pancreatic protection is not due to an anti-adipogenic, antilipemic, or systemic anti-inflammatory pathway, but the effect could be secondary to a specific mechanism involved in both pathological processes. Vascular endothelial dysfunction is a hallmark of most conditions associated with diabetes, as well as atherosclerosis. Recent studies show that hyperglycemia and atherosclerosis share many common mechanisms, such as endothelial activation and inflammation, mitochondrial oxidative stress, changes in extracellular matrix components, and disruption of cellular defense systems. As a result of diabetes, ROS production increases and antioxidant activity decreases, factors that are also closely linked to atherosclerosis [38]. 

Doxycycline can molecularly interfere with various common mechanism factors between hyperglycemia and atherosclerosis. Nuclear factor κB (NF-κB) is a transcription factor that triggers hyperglycemia and induces deleterious effects on endothelial function. In recent studies, doxycycline has been found to act on T-cell lymphoma and breast cancer cell lines by inhibiting the activation of NF-κB [40,41,42]. The inhibition of this protein may explain how doxycycline prevents the genesis of atherosclerosis and glucose metabolism disorders simultaneously. It would be necessary to verify this in future studies. On the other hand, TNF alpha production increases in individuals who eat a high-fat diet, inducing endothelial dysfunction, which could worsen atherosclerosis [43]. In agreement with the beneficial effect of doxycycline at the vascular level, it has been found that it is capable of inhibiting the expression of TNF alpha, along with other proinflammatory cytokines [44], which are phenomena associated with the genesis of diabetes [45] and atherosclerosis [46]. In addition to this, others studies mention that doxycycline is able to prevent testicular deterioration caused by the high-fat diet, which could suggest that doxycycline protects various endocrine organs [47]. All of the above is consistent with the results of the present study. 

In the search for new drugs and strategies to prevent and control diabetes, multiple therapeutic recommendations have been postulated. Some studies refer that the use of a ketogenic diet could improve metabolic aspects in the individual [48]. In this type of diet, the increase in ketone bodies helps to maintain the morpho-functional characteristics of pancreatic tissue associated with the homeostasis of blood glucose levels [48]. Considering the previous information, one perspective of the present study would be the evaluation of the effects of doxycycline in individuals with diabetes and other pathologies submitted to a ketogenic diet.

Doxycycline showed pharmacological characteristics that would allow its incorporation as adjuvant treatment in the prevention of diabetes and atherosclerosis in preclinical stages. An adverse effect of antibiotics on the host refers to damage to the microbiota, generating alterations in the metabolism of sugars and lipids. In this study, this factor was not controlled [49]. The gut microbiome plays an important role in extracting energy from food and inducing obesity [50]. It is conceivable that the effect of doxycycline on the microbiota could interfere with the body weight behavior of the animal model. Furthermore, it is important to note that a limitation of the non-transgenic animal model used is that it does not develop a significant increase in body weight from a chronically high-fat diet. Therefore, the effect of doxycycline on body weight in this model may not be adequate. It would be desirable to carry out future studies with other animal models and with a focus on preserving the intestinal microbiota. One of the strategies that could be established is the structural modification of doxycycline in its “A ring” (antimicrobial activity), reducing its action on the microbiota without losing its effects on other molecular targets [51]. Another interesting aspect to be addressed in future research is the effect of doxycycline on cardiac and physiological alterations related to mitochondrial function [52], since this was not evaluated in the present study. Additionally, epicardial fat was not evaluated, which constitutes an important factor to be analyzed in future studies. Finally, in the present study, the molecular mechanisms that prevent atherosclerosis and pancreatic histological damage were not analyzed, an aspect that needs to be studied in the future. 

## 5. Conclusions

The administration of doxycycline could be able to simultaneously prevents preclinical atherosclerosis and the loss of pancreatic islets and β cells in individuals chronically fed a high-fat diet, while possibly increasing the ability of β cells to release insulin into the blood. This could indicate a dual effect of doxycycline, as a protector of the morphology of the pancreatic islets and as an enhancer of the function of the β cells. Future studies should evaluate the potential use of doxycycline, or some derivative compounds of it, as possible adjuvants in the prevention of diabetes and cardiovascular diseases.

## Figures and Tables

**Figure 1 biomedicines-11-00717-f001:**
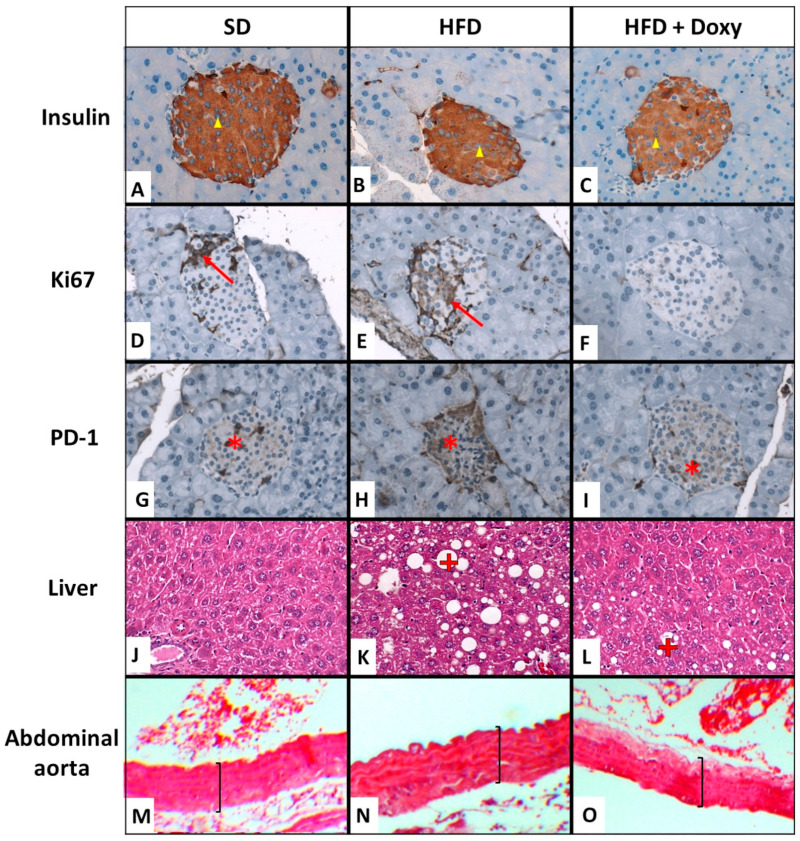
Histological sections of pancreatic tissue, liver, and abdominal aorta in the SD (standard diet), HFD (high-fat diet), and HDF + Doxy (high-fat diet + doxycycline) groups. (**A**–**I**): Islets of Langerhans from mice with SD (**A**,**D**,**G**), HFD (**B**,**E**,**H**), and HFD + Doxy (**C**,**F**,**I**) at the end of the experiment. Sections were immunostained for insulin (**A**–**C**), Ki67 (**D**–**F**), or PD-1 (**G**–**I**). (**A**,**C**) showed no statistical difference in the number of conserved beta cells (arrowhead) with insulin expression (brown-brown), (*p* = 0.873). (**B**) shows smaller pancreatic islets and fewer β cells. (**D**–**F)** were immunostained with Ki67. (**D**,**E**) show a positive reaction to the Ki67 marker (red arrows). (**E**) shows greater cell proliferation positivity compared to (**D**,**F**) (*p* < 0.05, for both comparisons). (**F**) shows a negative response to the Ki67 marker. G-I show islets immunostained with the PD-1 marker. (**G**,**I**) show low expression of the apoptosis marker with respect to (**H**) (∗). (**J**–**L**) show H&E-stained (X200) sections of liver tissue. (**J**) shows normal liver tissue. (**K**,**L**) show fatty deposits (+) and moderate to mild inflammation. (**M**–**O**) show H&E-stained (X50) sections of the abdominal aorta. (**N**) presents a greater thickness of the aorta (]) compared to (**M**,**O**). Regarding (**M**,**O**), there is no difference in the thickness of the aorta wall.

**Figure 2 biomedicines-11-00717-f002:**
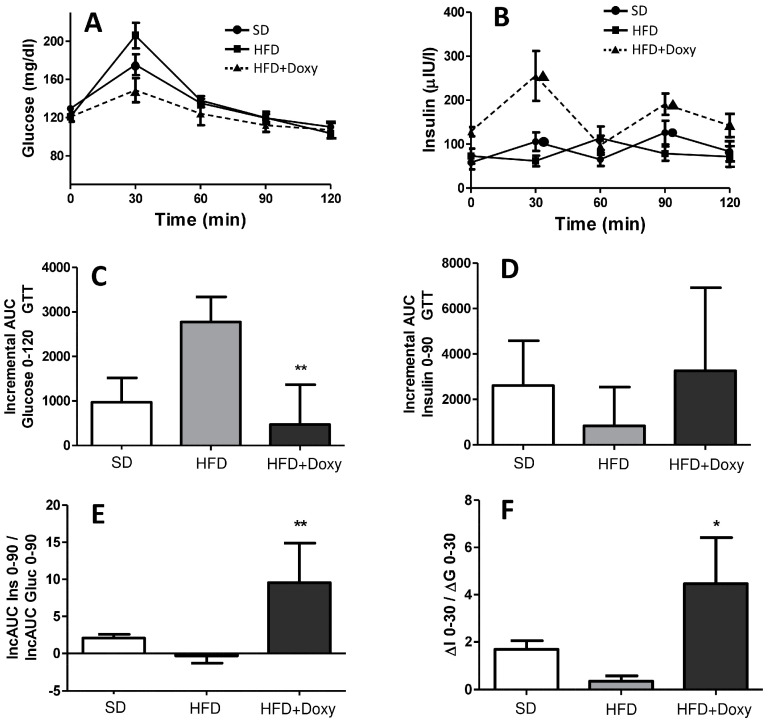
Serum glucose and insulin values in a glucose tolerance test (GTT). Data are shown as mean ± SD. ANOVA analyses show intergroup differences in (**A**–**C**,**F**) (*p* < 0.05 for all analyses). Changes in the mean blood glucose (**A**) and insulin (**B**) concentrations over 120 min. HFD + Doxy group at 30 min had a significantly lower glucose level compared with SD (*p* = 0.010) and HFD groups (*p* = 0.008). Additionally, the HFD group presented a significantly higher glucose value compared with the SD group at the same time point (*p* < 0.001). Meanwhile in the insulin comparison at 30 min, the HFD + Doxy group exhibits a significantly higher level when compared with the SD (*p* = 0.013) and HFD groups (*p* = 0.002), while the HFD and SD groups did not show differences. The incremental area under curve (gAUC) of the glucose measurements (**C**) show that the HFD + Doxy group had lower values that the HFD group (*p* < 0.001), but with similar values to that of the SD group (*p* = 0.520). The incremental area under curve (iAUC) of the insulin measurements (**D**) shows no statistical difference between the groups. After adjusting insulin secretion for the incremental AUC of glucose (**E**: iAUC/gAUC 0–90min ratio), elevated values are clearly shown in the HFD + Doxy group, compared to HFD (** *p* = 0.013), but similar values are seen when compared with the SD group (*p* = 0.147). ** Indicates statistically significant difference between HFD + Doxy and SD group. The β cell function index (**F**, ratio ∆Insulin0–30/∆Glucose0–30) indicates that the group with doxycycline had a better secretory function of the beta cell than the HFD and SD groups (*p* = 0.001, and *p* = 0.010, respectively). * Indicates statistically significant difference between HFD + Doxy and the rest of the groups (SD and HFD).

**Table 1 biomedicines-11-00717-t001:** Comparison of phenotypes and serum biochemical profiles between healthy mice (SD), fed with a high-fat diet for six months without doxycycline (HFD), and those with a high-fat diet and doxycycline (HFD + Doxy).

Parameter	SD	HFD	HFD + Doxy	*p*
Initial food intake (g) ^c^	2.91 ± 0.69	3.16 ± 0.56	2.89 ± 1.21	0.670
Final food intake (g) ^d^	6.86 ± 0.54	7.06 ± 0.94	6.70 ± 0.74	0.475
Total cholesterol (mg/dL)	138 ± 21	156 ± 51	144 ± 34	0.438
Tryglicerids (mg/dL)	224 ± 76	182 ± 49	191 ± 71	0.270
Glucose (mg/dL)	123.4 ± 11	119.3 ± 11	120.6 ± 16	0.276
Insulin (mU/L)	44 ± 30	79 ± 53	122 ± 24	0.020
Body weight (g)	31.5 ± 1.6	32.9 ± 2.2	31.9 ± 2.7	0.286
Epididymal fat (g)	0.8 ± 0.3	0.12 ± 0.3	0.18 ± 0.7	<0.001
Liver/Body weight ratio	0.046 ± 0.006	0.055 ± 0.006	0.058 ± 0.007	<0.001
Steatosis	0.4 ± 0.5	2.2 ± 0.4	1.8 ± 0.9	<0.001 ^b^
Liver inflammation	0.3 ± 0.6	1.5 ± 0.5	1.0 ± 0.3	<0.001 ^b^
ALT (U/L)	144 ± 21	190 ± 58	171 ± 35	0.030
AST (U/L)	1001 ± 166	933 ± 332	1128 ± 338	0.244
Width of abdominal aorta (µm)	62.9 ± 11	76.4 ± 18	60.7 ± 12	<0.001
Number of islets (per mm^2^)	7.5 (6–9.5)	4 (3–10)	9 (8–12)	0.030 ^b^
Size of islets µm^2^	8537 (3518–16,555)	7801 (5406–13,428)	7171 (3435–14,121)	0.959 ^b^
Islet size distribution				
Small Medium Large	55.3% 31.9% 12.8%	63.2% 36.8% 0.0%	51.6% 37.5% 10.9%	0.528 ^b^
Number of β cells (per mm^2^)	1179 (622–1990)	466 (264–932)	1202 (819–1341)	0.002 ^b^
Single-cell area β µm^2^	95 (91–105)	114 (95–120)	104 (100–111)	0.002 ^b^
Beta cell cytoplasmatic area µm^2^	76.4 (69–85)	93.7 (78–99)	84.1 (79–89)	0.002 ^b^
Percentage of positive area per islet				
Insulin Ki67 PD-1	66.5 (61.3–70.7) 1.6 (0.0–3.3) 3.7 (0.0–9.6)	66.4 (59.0–72.0) 6.2 (3.1–13.3) 6.6 (3.6–15.1)	65.2 (53.4–70.4) 0.0 (0.0–4.4) 3.5 (0.0–8.3)	0.554 ^b^ <0.001 ^b^ 0.148 ^b^

The values of biochemical parameters are expressed as the means ± SD or median (percentile 25–75). Number of islets (count per mm^2^ of pancreas). *p*: result of statistical analysis with ANOVA test, except for values marked with ^b^, for which Kruskal–Wallis test was used. ^c^ Food intake media at first week of the study, ^d^ Food intake media at week 24 of the study.

**Table 2 biomedicines-11-00717-t002:** Statistical comparison (post hoc) of various parameters between healthy (SD), high-fat diet (HFD), and high-fat diet and doxycycline (HFD + Doxy) mice groups.

Parameter	SD vs. HFD	SD vs. HFD + Doxy	HFD vs. HFD + Doxy
Insulin (mU/L)	0.285 ^a^	0.019 ^a^	0.309 ^a^
ALT (U/L)	0.035 ^a^	0.226 ^a^	0.790 ^a^
Liver/Body weight ratio	0.013 ^a^	<0.001 ^a^	0.802 ^a^
Epididymal fat (g)	0.205 ^a^	0.000 ^a^	0.015 ^a^
Width of abdominal aorta (µm)	<0.001 ^a^	0.720 ^a^	<0.001 ^a^
Liver inflammation	<0.001 ^b^	0.002 ^b^	0.120 ^b^
Steatosis	<0.001 ^b^	<0.001 ^b^	0.403 ^b^
Number of islets	0.241 ^b^	0.082 ^b^	0.047 ^b^
Number of beta cells	0.036 ^b^	0.873 ^b^	0.030 ^b^
Area of beta cells	0.047 ^b^	0.042 ^b^	0.497 ^b^
Cytoplasmic area of beta cell	0.031 ^b^	0.059 ^b^	0.418 ^b^
Ki67-positive area per islet	0.001 ^b^	0.223 ^b^	<0.001 ^b^

The values are expressed as *p* values. ^a^ Dunnett’s post hoc test; ^b^ Mann–Whitney U post hoc test.

## Data Availability

All relevant data appears in the paper. The datasets used and/or analyzed in the current study are available from the corresponding author upon reasonable request.

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
