# Peer review of "Doxycycline Prevents Preclinical Atherosclerosis, Pancreatic Islet Loss and Improves Insulin Secretion after Glycemic Stimulation: Preclinical Study in Individuals with a High-Fat Diet"

_biomedicines, 2023, doi:10.3390/biomedicines11030717_

Round 1

Reviewer 1 Report

In the manuscript entitled “Doxycycline prevents preclinical atherosclerosis, pancreatic islet loss and improves insulin secretion after glycemic stimulation: Preclinical study in individuals with a high-fat diet” the author evaluated the effect of Doxycycline on HFD diet-induced metabolic disorders in BALB/c mice and concluded that administration of doxycycline prevents atherosclerosis and the loss of pancreatic islets and B-cells fed with HFD. These results are preliminary, and the conclusions are overstated. The paper also suffers from many other problems, which detract from the overall quality.   

  1. In the “Materials and Methods”, more details about the high-fat diet and doxycycline, including the percentage of fat and catalog numbers are required.  
  2. There are no differences in the serum TG, cholesterol, insulin, and glucose levels between SD and HFD groups, especially the comparable body weight between SD and HFD groups which indicates the HFD mice models were not successful in this manuscript   
  3. Does doxycycline administration have any effect on the food intake in the HFD+Doxy group?  
  4. The pancreas IHC staining images (anti-insulin; anti-PD1; anti-Ki67) were not shown in the manuscript, and the results about the changes in pancreatic morphology, pancreatic islets proliferation, and apoptosis are not convincing.  
  5. The liver and aorta histopathological staining results were not shown in the manuscript, the results about the changes of steatosis and inflammatory infiltration are not convincing.  
  6. What’s Figure 1 represent? The figure legend and quantification for Figure 1 are required.  
  7. For Figure 2, the significant differences between different groups are required.  
  8. More detailed information and evidence are required to get the conclusion described in the manuscript.  

Author Response

Comment: In the “Materials and Methods”, more details about the high-fat diet and doxycycline, including the percentage of fat and catalog numbers are required.

Answer: Thank for your observation, the details of the high fat diet and doxycycline are now described in more detail in the methodology section, as follows:

“Doxycycline was purchased from Sigma Aldrich (D9891, MO, USA). A standard diet was used (2018S Tekland Global 18% Protein Rodent Diet, Harlan®, USA) containing 18.6% protein, 46.9% carbohydrate and 13.2% fat. The high-fat diet used (TD.02028 Atherogenic Rodent Diet, Harlan®, USA) contains protein 17.3%, carbohydrate 46.9%, and fat 21.2%.”

Comment:-There are no differences in the serum TG, cholesterol, insulin, and glucose levels between SD and HFD groups, especially the comparable body weight between SD and HFD groups which indicates the HFD mice models were not successful in this manuscript  

Answer: Thank you, attending this important observation, It should be noticed that the non-transgenic mice model fed with a high fat diet used in these experiments allowed the evaluation of preclinical stages of diabetes and atherosclerosis. It is important to mention that in preclinical stages of metabolic disorders, the fasting serum parameters of lipids, glucose and insulin may not present alterations, so it is necessary to evaluate other more predictive or sensitive test to establish differences between individuals. The following references were included:

  • doi:10.3923/ajava.2014.202.210.
  • doi:10.1016/j.ijbiomac.2021.08.003.
  • doi:10.1016/S0140-6736(20)32511-3.

Comment:-Does doxycycline administration have any effect on the food intake in the HFD+Doxy group?

Answer: Excellent question. In our results, we included a short sentence referring to food intake "doxycycline administration in the high-fat diet groups did not lead to significant changes in the amount of product consumed (data not shown)". Other studies have found similar behaviours under the effect of doxycycline. Ref. 10.2147/DMSO.S292264

Comment:-The pancreas IHC staining images (anti-insulin; anti-PD1; anti-Ki67) were not shown in the manuscript, and the results about the changes in pancreatic morphology, pancreatic islets proliferation, and apoptosis are not convincing.

Answer: Thank you for the observation. The IHC staining images of the pancreas (insulin, PD1 and Ki67) were integrated into Figure 1 of this manuscript, presenting more clearly the results about the changes in pancreatic morphology and pancreatic islets proliferation.

Comment:-Thank you. The liver and aorta histopathological staining results were not shown in the manuscript, the results about the changes of steatosis and inflammatory infiltration are not convincing.  What’s Figure 1 represent? The figure legend and quantification for Figure 1 are required. 

Answer: Thanks for the observation. The legend of Figure 1 describes the images shown in more detail. The figure legend was included in the manuscript as follows:

Figure 1. Histological sections of pancreatic tissue, liver and abdominal aorta in the SD (standard diet), HFD (high-fat diet) and HDF+Doxy (high-fat diet + doxycycline) groups. Figure A-I: Islets of Langerhans from mice with SD (A, D and G), HFD (B, E and H) and HFD+Doxy (C, F and I) at the end of the experiment. Sections were immunostained for insulin (A, B, and C), Ki67 (D, E, and F), or PD-1 (G, H, and I). A and C showed no statistical difference in the number of conserved beta cells (arrowhead) with insulin expression (brown-brown), (p= 0.873). B shows smaller pancreatic islets and fewer β cells. D-F were immunostained with Ki67. D and E show a positive reaction to the Ki67 marker (red arrows). E shows greater cell proliferation positivity compared to D and F (p<0.05). F shows a negative response to the Ki67 marker. G-I show islets immunostained with the PD-1 marker. G and I show low expression of the apoptosis marker with respect to H (*). J-L show H&E-stained (X200) sections of liver tissue. J shows normal liver tissue. K and L show fatty deposits (+) and moderate to mild inflammation. M-O show H&E-stained (X50) sections of the abdominal aorta. N presents a greater thickness of the aorta (]) compared to M and O. Regarding M and O, there is no difference in the thickness of the aorta.

Comment:-For Figure 2, the significant differences between different groups are required.

Answer: Thanks for the observation. The legend of the figure was corrected, helping to improve the interpretation of the image. The figure legend was included in the manuscript as follows:

Figure 2. Serum glucose and insuline values in a glucose tolerance test (GTT). Data are shown as mean ± SD. ANOVA analyses show intergroup differences in A, B, C and F (P<0.05 for all analyses). Changes in the mean blood glucose (A) and insuline (B) concentrations over 120 minutes. HFD + Doxy group at 30 min had a significantly lower glucose level compared with SD (p= 0.010) and HFD groups (p = 0.008). Also, the HFD group presented significantly higher glucose value compared with SD group in the same time point (p < 0.001). Meanwhile in the insuline comparison at 30 min, the HFD + Doxy group exhibits a significantly higher level at compared with SD (p = 0.013) and HFD groups (p = 0.002), while the HFD and SD groups did not show differences. The incremental area under curve (gAUC) of the glucose measurements (C) show that the HFD+Doxy group had lower values that the HFD group (p < 0.001), but with similar values that SD group (p = 0.520). After adjusting insulin secretion for the incremental AUC of glucose (E: iAUC/gAUC 0-90min ratio), elevated values are clearly shown in the HFD+Doxy group, compared to HFD (**p = 0.013), but shows similar values at compared with SD group (p = 0.147). The β cell function index (F, ratio ∆Insulin0–30/∆Glucose0–30) indicates that the group with doxycycline had a better secretory function of the beta cell than the HFD and SD groups (p = 0.001, and p = 0.010, respectively). 

Comment:-More detailed information and evidence are required to get the conclusion described in the manuscript.

Answer: Following the reviewer's suggestion. It should be noted that the categorical sentences were corrected both in the conclusion and discussion, helping to specify the findings of this study.

Reviewer 2 Report

The authors studied effect of Doxycycline antibiotic on the preclinical atherosclerosis, pancreatic islet loss and insulin secretion in a high-fat-diet preclinical mouse model.

This research presents interesting findings on the anti-atherosclerotic effect of the antibiotic, doxycycline as well as its effect in improving insulin secretion after glycemic stimulation. They also used good research plan and used several biochemical, histological and immuno-histochemical approaches and methods to show the anti-atherosclerotic effect and GTT- improving effect of doxycycline in a high-fat diet model of hyperlipidemia and HFD-induced insulin resistance/T2D.  

These findings fill some gaps in the current literature of this point, which is still underdeveloped. However, there is serious key issue regarding the experimental design and the dose of the antibiotic that was missed in this study.

It is well-established in the literature, long time ago, that chronic use of antibiotic has adverse effect on the intestinal microbiota, depressing the beneficial bacteria, that would otherwise improve metabolism, digestion, anti-inflammatory and antilipidemic effects in the host intestine. There is great cumulative evidence supporting this argument. Just an example: see Ahn Y. et al. Antibiotics 2021, 10(8), 886; https://doi.org/10.3390/antibiotics10080886. Impact of Chronic Tetracycline Exposure on Human Intestinal Microbiota in a Continuous Flow Bioreactor Model.

According to this paper: the European Medicines Agency (EMA) determined the ADI value for tetracycline to be 0.18 mg/60 kg bw/day, equivalent to 3 μg/kg bw/day, based on a measured resistance endpoint [20,21] (pasted, cited from Ahn Y. et al. Antibiotics 2021, 10(8), 886, mentioned above. Again, there is tons of evidence on this point, this paper just a relevant example. Further, microflora is a hot topic/player, where most research on drugs work on.

This point was NOT controlled in this study, under review herein (e.g., this effect can be encountered and controlled by adding probiotic water or food or a pre-determined dose of CFUs by oral gavage every 1-2 weeks to compensate this side effect of the antibiotic, and doing the same for the HFD-saline group and standard diet.   

Given the dose of 10 mg/kg body weight is several-fold more than what most studies reported to avoid affecting microbiota (unlike using antibiotic in infection model for 2 weeks, such dose of 10 mg/kg/day may be fine). Further, 6 months age period roughly equals 20 years or so in mice life. This implies that such chronic use of the antibiotic must be controlled in its side effects on intestinal flora.

This may explain, at least partially, the possibility why the doxycycline did not affect steatosis, lipidemia and insulin resistance in the presumably non-fasting glucose, with higher insulin in HFD + doxycycline group vs. St. Diet, and numerically HFD+ doxycycline vs. HFD in table 1 (not sig. apparently due to variation), when glucose level is similar among the groups.  

However, the Only possible solution for this critical point to be addressed and current manuscript to be published, in my scientific views, is to incorporate this point in discussion as 1) stating comment on the possibility behind the “no effect of doxycycline on lipedema and steatosis is the adverse effect of antibiotic on the microbiota as I indicated above. 2) adding another limitation at the end of discussion, stating that the adverse effect of antibiotic, doxycycline in microbiota was not controlled and encountered, which is worth further future studies to explore or verify the prospective effect of doxycycline in insulin resistance, lipidemia, steatosis …etc. This way of drafting presents a clear, comprehensive, and more informative message to the readers, learners, and researchers.

A sub-title of limitations may be suitable.

Other comments and minor edits:

1-      Line 57: due to its inhibition of protein synthesis (add: to).

2-      Line 82:  in mice model of hyperlipidemia and hyperglycemia or model of insulin resistance (type II diabetes). Add definition words of the model.

3-      Line 84: better to re-write as: The present research work was conducted to assess.

4-      Line 97, 98: It does not read clearly.  

   Either the authors want to say: this model previously demonstrated to produce steatohepatitis, preclinic atherosclerosis and metabolic alterations [10]. In this case,  the cited reference (10) in line 98  is not showing that this HFD Atherogenic Roent Diet, Harlan® , USA) induces atherosclerosis and metabolic alterations. This published report (ref. 10) shows the antioxidant properties of doxycycline in treatment of cardiovascular disease.

5-      Line 216: legend of table 2: It has to be clearly written (p values of statistical comparison tests…etc.

Methods: Line 176 (Biochemical analysis): If slaughter of mice was done without prior removal of food (e.g., 6 or 12 hrs fasting), it is clearer to state that in method and/or legend of table 1 (before slaughter, blood samples were collected from non-fasting mice), just to confirm to the reader hat certainly these were normal OR food pellets were removed 6 or 8 hrs before the time of the experiment 9if this was the case).

Discussion:

Page 9, line 323: pasting from text: Doxycycline is an inhibitor of NF-κB [27]. In this reference doxycycline inhibited NF-kB in in two sub-types of T cell lymphoma cell line. To put the reference in its scientific context, more appropriate to cite this way, stated in the upper line. It is sound to cite it, but without exceeding its limit of the meaning for accuracy. citing any further in in vitro or in vivo published studies, if any, where doxycycline inhibited NF-kB can be added.

Other limitations/future directions:

Refining and adopting mouse model of atherosclerosis, separately from another refined HFD-induced T2D mouse model, can be a valid strategy, worth searching and reviewing it for further and more clear-cut on the doxycycline’s effect on atherosclerosis and T2D.

Consider visiting this review reference: Recent Pat Cardiovasc Drug Discovery.

. 2011 Jan;6(1):42-54. Effect of doxycycline on atherosclerosis: from bench to bedside, doi: 10.2174/157489011794578419.

Considerations for Future studies: rationale, exp. designed and planned controls should also consider this reference for adverse effects of doxycycline on cardiac mitochondrial and contractile function, at least preparation of defence in discussion, if it diverges from a strong potential beneficial effect of this antibiotic.

The Antibiotic Doxycycline Impairs Cardiac Mitochondrial and Contractile Function

Rob C. I. Wüst, Bram F. Coolen, [...], and Riekelt H. Houtkooper. Int J Mol Sci Apr; 22(8): 4100.

Author Response

Comment:

This research presents interesting findings on the anti-atherosclerotic effect of the antibiotic, doxycycline as well as its effect in improving insulin secretion after glycemic stimulation. They also used good research plan and used several biochemical, histological and immuno-histochemical approaches and methods to show the anti-atherosclerotic effect and GTT- improving effect of doxycycline in a high-fat diet model of hyperlipidemia and HFD-induced insulin resistance/T2D. 

These findings fill some gaps in the current literature of this point, which is still underdeveloped. However, there is serious key issue regarding the experimental design and the dose of the antibiotic that was missed in this study.

It is well-established in the literature, long time ago, that chronic use of antibiotic has adverse effect on the intestinal microbiota, depressing the beneficial bacteria, that would otherwise improve metabolism, digestion, anti-inflammatory and antilipidemic effects in the host intestine. There is great cumulative evidence supporting this argument. Just an example: see Ahn Y. et al. Antibiotics 2021, 10(8), 886; https://doi.org/10.3390/antibiotics10080886. Impact of Chronic Tetracycline Exposure on Human Intestinal Microbiota in a Continuous Flow Bioreactor Model.

According to this paper: the European Medicines Agency (EMA) determined the ADI value for tetracycline to be 0.18 mg/60 kg bw/day, equivalent to 3 μg/kg bw/day, based on a measured resistance endpoint [20,21] (pasted, cited from Ahn Y. et al. Antibiotics 2021, 10(8), 886, mentioned above. Again, there is tons of evidence on this point, this paper just a relevant example. Further, microflora is a hot topic/player, where most research on drugs work on.

This point was NOT controlled in this study, under review herein (e.g., this effect can be encountered and controlled by adding probiotic water or food or a pre-determined dose of CFUs by oral gavage every 1-2 weeks to compensate this side effect of the antibiotic, and doing the same for the HFD-saline group and standard diet.  

Given the dose of 10 mg/kg body weight is several-fold more than what most studies reported to avoid affecting microbiota (unlike using antibiotic in infection model for 2 weeks, such dose of 10 mg/kg/day may be fine). Further, 6 months age period roughly equals 20 years or so in mice life. This implies that such chronic use of the antibiotic must be controlled in its side effects on intestinal flora.

This may explain, at least partially, the possibility why the doxycycline did not affect steatosis, lipidemia and insulin resistance in the presumably non-fasting glucose, with higher insulin in HFD + doxycycline group vs. St. Diet, and numerically HFD+ doxycycline vs. HFD in table 1 (not sig. apparently due to variation), when glucose level is similar among the groups. 

However, the Only possible solution for this critical point to be addressed and current manuscript to be published, in my scientific views, is to incorporate this point in discussion as 1) stating comment on the possibility behind the “no effect of doxycycline on lipedema and steatosis is the adverse effect of antibiotic on the microbiota as I indicated above. 2) adding another limitation at the end of discussion, stating that the adverse effect of antibiotic, doxycycline in microbiota was not controlled and encountered, which is worth further future studies to explore or verify the prospective effect of doxycycline in insulin resistance, lipidemia, steatosis …etc. This way of drafting presents a clear, comprehensive, and more informative message to the readers, learners, and researchers.

However, the Only possible solution for this critical point to be addressed and current manuscript to be published, in my scientific views, is to incorporate this point in discussion as 1) stating comment on the possibility behind the “no effect of doxycycline on lipedema and steatosis is the adverse effect of antibiotic on the microbiota as I indicated above. 2) adding another limitation at the end of discussion, stating that the adverse effect of antibiotic, doxycycline in microbiota was not controlled and encountered, which is worth further future studies to explore or verify the prospective effect of doxycycline in insulin resistance, lipidemia, steatosis …etc. This way of drafting presents a clear, comprehensive, and more informative message to the readers, learners, and researchers.

Answer: Thank you for your detailed and accurate comment. You address a highly relevant topic and we appreciate you even suggesting how to solve the limitation of our manuscript on this topic. Thank you so much. This observation was incorporated in the limitations section, highlighting its importance for future research.

“Doxycycline showed pharmacological characteristics that would allow its incorporation as adjuvant treatment in the prevention of diabetes and atherosclerosis in preclinical stages. Additionally, with the information collected, new molecules derived from doxycycline could be established that would improve its effects. An adverse effect of antibiotics on the host refers to damage to the microbiota, generating alterations in the metabolism of sugars and lipids. In this study, this factor was not controlled, so it could open new lines of research where it preserves the microbiota [45]. doi:10.3390/antibiotics1010001.”

Comment: Line 57: due to its inhibition of protein synthesis (add: to).

Answer: Thanks for the observation, the preposition was incorporated

Comment: Line 82:  in mice model of hyperlipidemia and hyperglycemia or model of insulin resistance (type II diabetes). Add definition words of the model.

Answer: Thanks for the observation, the statement was rewritten, and the name of the mouse model (non-transgenic mice model) was incorporated.

Comment: Line 84: better to re-write as: The present research work was conducted to assess.

Answer: Thank you for the observation. The sentence was rewritten.

Comment: Line 97, 98: It does not read clearly.  Either the authors want to say: this model previously demonstrated to produce steatohepatitis, preclinic atherosclerosis and metabolic alterations [10]. In this case, the cited reference (10) in line 98 is not showing that this HFD Atherogenic Roent Diet, Harlan®, USA) induces atherosclerosis and metabolic alterations. This published report (ref. 10) shows the antioxidant properties of doxycycline in treatment of cardiovascular disease.

Answer: Thank you for your observation, we made a change in the reference list incorporating the following references that would support the use of a high fat diet to generate hepatic steatosis.

Getz, G.S.; Reardon, C.A. Diet and Murine Atherosclerosis. ATVB 2006, 26, 242–249, doi:10.1161/01.ATV.0000201071.49029.17.

VanSaun, M.N.; Lee, I.K.; Washington, M.K.; Matrisian, L.; Gorden, D.L. High Fat Diet Induced Hepatic Steatosis Establish-es a Permissive Microenvironment for Colorectal Metastases and Promotes Primary Dysplasia in a Murine Model. The American Journal of Pathology 2009, 175, 355–364, doi:10.2353/ajpath.2009.080703.

Garcìa-Rivera, A.; Madrigal-Perez, V.M.; Rodriguez-Hernandez, A.; Martinez-Martinez, R.; Martinez-Fierro, M.L.; Soriano-Hernandez, A.D.; Galvan-Salazar, H.R.; Gonzalez-Aivarez, R.; Valdez-Yelazquez, L.L.; Espinoza-Gómez, F.; et al. A Simple and Low-Cost Experimental Mouse Model for the Simultaneous Study of Steatohepatitis and Preclinical Atherosclerosis. Asian Journal of Animal and Veterinary Advances 2014, 9, 202–210, doi:10.3923/ajava.2014.202.210.

Comment: Line 216: legend of table 2: It has to be clearly written (p values of statistical comparison tests…etc

Answer: Thanks for the observation, the legend of table 2 was completed.

Comment: Methods: Line 176 (Biochemical analysis): If slaughter of mice was done without prior removal of food (e.g., 6 or 12 hrs fasting), it is clearer to state that in method and/or legend of table 1 (before slaughter, blood samples were collected from non-fasting mice), just to confirm to the reader hat certainly these were normal OR food pellets were removed 6 or 8 hrs before the time of the experiment if this was the case)

Answer: Thanks for your observation. The blood sample collection is now described more fully in the methodology section. The sentence was rewritten in this section: “Before the slaughter blood samples were collected from the mice with 6 hours of fasting to measure serum lipids (triglycerides, total cholesterol) and liver enzymes (ALT, AST)”

Comment: Page 9, line 323: pasting from text: Doxycycline is an inhibitor of NF-κB [27]. In this reference doxycycline inhibited NF-kB in in two sub-types of T cell lymphoma cell line. To put the reference in its scientific context, more appropriate to cite this way, stated in the upper line. It is sound to cite it, but without exceeding its limit of the meaning for accuracy. citing any further in in vitro or in vivo published studies, if any, where doxycycline inhibited NF-kB can be added.

Answer: Thanks for the observation. The statement was rewritten in discussion section in order to reference it in a more appropriate context and the suggested reference was incorporated into the manuscript.

“Doxycycline can molecularly interfere with various common mechanism factors between hyperglycemia and atherosclerosis. Nuclear factor κB (NF-κB), is a transcription factor that triggers hyperglycemia andinduce deleterious effects on endothelial function. In recent studies, doxycycline has been found to act on T-cell lymphoma and breast cancer cell lines by inhibiting the activation of NF-κB. The inhibition of this protein may explain how doxycycline prevents the genesis of atherosclerosis and glucose metabolism disorders simultaneously.”

Comment: Refining and adopting mouse model of atherosclerosis, separately from another refined HFD-induced T2D mouse model, can be a valid strategy, worth searching and reviewing it for further and more clear-cut on the doxycycline’s effect on atherosclerosis and T2D. Consider visiting this review reference: Recent Pat Cardiovasc Drug Discovery. 2011 Jan;6(1):42-54. Effect of doxycycline on atherosclerosis: from bench to bedside, doi: 10.2174/157489011794578419.

Answer: Thanks for your comment. The suggested reference revision helped us improve the discussion of the manuscript and was incorporated into the manuscript.

Comment: Considerations for Future studies: rationale, exp. designed and planned controls should also consider this reference for adverse effects of doxycycline on cardiac mitochondrial and contractile function, at least preparation of defence in discussion, if it diverges from a strong potential beneficial effect of this antibiotic. The Antibiotic Doxycycline Impairs Cardiac Mitochondrial and Contractile Function Rob C. I. Wüst, Bram F. Coolen, [...], and Riekelt H. Houtkooper. Int J Mol Sci Apr; 22(8): 4100.

Answer: Thanks for your comment. The suggested reference helped us to improve the discussion of the manuscript, for which it was incorporated into the manuscript in that section, allowing it to be considered in future studies.

Reviewer 3 Report

The article demonstrates the impact of doxyclycline administration on the insulin function in rodents and it provides new applications of the drug. The findings are impressive given the fact that it demonstrates pleiotropic effects of an antibiotic. The metodology of the study is accurate and the design is well balanced serving to demonstrate the idea of the study. The discussion chapter can be improved please discuss the impact on ketone bodies production on the beta cell function in the setting of a high fat diet. please consult Implicating the effect of ketogenic diet as a preventive measure to obesity and diabetes mellitus. Life Sci. 2021 Jan 1;264:118661. doi: 10.1016/j.lfs.2020.118661. Epub 2020 Oct 26. PMID: 33121986. Minor revision.

Author Response

Comment: The article demonstrates the impact of doxyclycline administration on the insulin function in rodents and it provides new applications of the drug. The findings are impressive given the fact that it demonstrates pleiotropic effects of an antibiotic. The methodology of the study is accurate and the design is well balanced serving to demonstrate the idea of the study. The discussion chapter can be improved please discuss the impact on ketone bodies production on the beta cell function in the setting of a high fat diet. please consult Implicating the effect of ketogenic diet as a preventive measure to obesity and diabetes mellitus. Life Sci. 2021 Jan 1;264:118661. doi: 10.1016/j.lfs.2020.118661. Epub 2020 Oct 26. PMID: 33121986. Minor revision.

Answer: Thanks for the comment, the suggested reference was included in the discussion, as shown below, allowing it to be considered in future studies.

“In the search for new drugs and strategies to prevent and control diabetes, multiple therapeutic recommendations have been postulated. Some studies refer that the use of a ketogenic diet could improve metabolic aspects in the individual. In this type of diet, the increase in ketone bodies helps to maintain the morphofunctional characteristics of pancreatic tissue associated with the homeostasis of blood glucose levels. Considering the previous information, one perspective of the present study would be the evaluation of the effects of doxycycline in individuals with diabetes and other pathologies submitted to a ketogenic diet.”

Kumar, S.; Behl, T.; Sachdeva, M.; Sehgal, A.; Kumari, S.; Kumar, A.; Kaur, G.; Yadav, H.N.; Bungau, S. Implicating the Effect of Ketogenic Diet as a Preventive Measure to Obesity and Diabetes Mellitus. Life Sciences 2021, 264, 118661, doi:10.1016/j.lfs.2020.118661.

Reviewer 4 Report

I read the paper “Doxycycline prevents preclinical atherosclerosis, pancreatic islet loss and improves insulin secretion after glycemic stimulation: Preclinical study in individuals with a high-fat dietby Alejandrina Rodriguez Hernandez et al.

The manuscript is quite easy to ready. Statistical analysis is well performed.

Comments:

1.      There are a lot of typo errors throughout the manuscript. Please revise.

2.      The authors have reported in the title the role of doxycycline in preclinical atherosclerosis. I suggest reporting some information in the introduction section. In particular, it can be added as follow “Diabetes confers about a two-fold excess risk of coronary heart disease, stroke, and death due to other vascular causes. Thus the need of knowledge improvement in finding new therapeutic approaches in atherosclerosis prevention and treatment” (doi: 10.1186/s12933-022-01674-7).

3.      Do you have data on epicardial fat? It would have been interesting to also evaluate this point.

4.      Do you think that doxycycline use could be of any help in future for the treatment of diabetes, or do you think it will help to improve knowledge with consequent development of newer drugs? This issue because I am not certain that this could be of direct help.

5. High fat diet have proven to induce an increase in TNFalfa production with increased endothelial dysfunction, which might worsen atherosclerosis (doi: 10.1016/j.numecd.2005.11.014). The use of doxycyclin have also improved atherosclerosis. Please comment it.

Author Response

Comment: There are a lot of typo errors throughout the manuscript. Please revise.

Answer: Thanks for the observation, the document was reviewed and corrected in the text and format. Also, the manuscript was reviewed by a professional English-speaking editor. We attach the English language reviewer curriculum at the end of the cover letter PDF file.

Comment: The authors have reported in the title the role of doxycycline in preclinical atherosclerosis. I suggest reporting some information in the introduction section. In particular, it can be added as follow “Diabetes confers about a two-fold excess risk of coronary heart disease, stroke, and death due to other vascular causes. Thus, the need of knowledge improvement in finding new therapeutic approaches in atherosclerosis prevention and treatment” (doi: 10.1186/s12933-022-01674-7).

Answer: Thank you for the observation and the important suggestion. The phrase was incorporated into the introduction section, and the reference was included in this manuscript.

“Diabetes confers about a two-fold excess risk of coronary heart disease, stroke, and death due to other vascular causes. It has been demonstrated that doxycycline administration could reduce the formation of atherosclerosis or its pathophysiologic process in different mice models generated through mechanic endothelial damage, bacterial infection, or high fat diet.”

Sasso, F.C.; Simeon, V.; Galiero, R.; Caturano, A.; De Nicola, L.; Chiodini, P.; Rinaldi, L.; Salvatore, T.; Lettieri, M.; Nevola, R.; et al. The Number of Risk Factors Not at Target Is Associated with Cardiovascular Risk in a Type 2 Diabetic Population with Albuminuria in {Primary Cardiovascular Prevention. Post-Hoc Analysis of the NID-2 Trial. Cardiovasc Diabetol 2022, 21, 235, doi:10.1186/s12933-022-01674-7.

Comment: Do you have data on epicardial fat? It would have been interesting to also evaluate this point.

Answer: Thank you for your observation. Epicardial fat was not measured in this study. However, its importance is discussed as a limitation in the discussion section, in order to be considered in future studies.

Comment: Do you think that doxycycline use could be of any help in future for the treatment of diabetes, or do you think it will help to improve knowledge with consequent development of newer drugs? This issue because I am not certain that this could be of direct help.

Answer: Thanks for the observation, in this study doxycycline showed benefits on some metabolic variables, which could support its future incorporation as an adjuvant treatment in metabolic disorders, which is described in the discussion. The modification of the doxycycline molecule could improve the observed pharmacological activity. However, additional studies are required to verify these properties.

Comment: High fat diet have proven to induce an increase in TNF alfa production with increased endothelial dysfunction, which might worsen atherosclerosis (doi: 10.1016/j.numecd.2005.11.014). The use of doxycycline have also improved atherosclerosis. Please comment it.

Answer: Thanks for the observation. Information regarding the link between TNF alpha and atheroscerosis and the use of doxycycline was included in the discussion section, as follows. Also, the suggested reference was included in the manuscript.

” On the other hand, TNF alpha production increases in individuals who eat a high-fat di-et, inducing endothelial dysfunction, which could worsen atherosclerosis [39]. In agreement with the beneficial effect of doxycycline at the vascular level, it has been found that it is capable of inhibiting the expression of TNF alpha, along with other proinflammatory cytokines”

Esposito, K.; Ciotola, M.; Sasso, F.C.; Cozzolino, D.; Saccomanno, F.; Assaloni, R.; Ceriello, A.; Giugliano, D. Effect of a Single High-Fat Meal on Endothelial Function in Patients with the Metabolic Syndrome: Role of Tumor Necrosis Factor-α. Nutrition, Metabolism and Cardiovascular Diseases 2007, 17, 274–279, doi:10.1016/j.numecd.2005.11.014

Round 2

Reviewer 1 Report

In the revised manuscript, the author addressed most of the questions raised in the previous version. However, there are still some lingering issues, as described in Comment 2, the author attempts to generate the HFD-induced obesity model, which differs from the genetic-based db/db model. however, the  comparable body weight after 6 months of both the chow diet and high-fat diet feeding raises questions about the success of the HFD model and credibility of other conclusions."

Additionally, for Comment 3, the response of "data not shown" is unacceptable and it would be better to provide adequate evidence to support the conclusion.

Author Response

Comment:  In the revised manuscript, the author addressed most of the questions raised in the previous version. However, there are still some lingering issues, as described in Comment 2, the author attempts to generate the HFD-induced obesity model, which differs from the genetic-based db/db model. however, the comparable body weight after 6 months of both the chow diet and high-fat diet feeding raises questions about the success of the HFD model and credibility of other conclusions."

Answer: Thank you very much for your comment. We consider it very pertinent to clarify this observation in the text. It is important to mention that the main objective of the animal model used is not to generate an increase in body weight, but to generate morphological and physiological changes typical of chronic-degenerative diseases that ultimately lead to glucose metabolism disorders and diseases associated with a diet rich in fat. This aspect has also been evidenced in previous studies with non-transgenic mouse models fed a high-fat diet. This has been made clear in the methodology section when describing the animal model, adding bibliographical references. In the "Discussion" section, the limitation of the animal model for analyzing the effect of doxycycline on body weight was added.

“In this study a model of preclinical atherosclerosis and non-alcoholic steatohepatitis was employed with genotype native mice (BALB/c). The mice were subjected to a high-fat diet in a period of 6 months inducing metabolic dysregulation [13]. This mouse model is ideal for evaluating aspect that intervenes in a slow development of atherosclerosis and non-alcoholic steatohepatitis in the initial stages [18]. This and other similar models of non-transgenic mice do not develop significant increases in body weight, but do develop morpho-histological and metabolic changes in the early stages of chronic diseases associated with high-fat diets[19–22]”.

  1. Garcìa-Rivera, A.; Madrigal-Perez, V.M.; Rodriguez-Hernandez, A.; Martinez-Martinez, R.; Martinez-Fierro, M.L.; Soriano-Hernandez, A.D.; Galvan-Salazar, H.R.; Gonzalez-Aivarez, R.; Valdez-Yelazquez, L.L.; Espinoza-Gómez, F.; et al. A Simple and Low-Cost Experimental Mouse Model for the Simultaneous Study of Steatohepatitis and Preclinical Atherosclerosis. Asian Journal of Animal and Veterinary Advances 2014, 9, 202–210, doi:10.3923/ajava.2014.202.210.
  2. Powell, E.E.; Wong, V.W.-S.; Rinella, M. Non-Alcoholic Fatty Liver Disease. The Lancet 2021, 397, 2212–2224, doi:10.1016/S0140-6736(20)32511-3.
  3. Nishikawa, S.; Yasoshima, A.; Doi, K.; Nakayama, H.; Uetsuka, K. Involvement of Sex, Strain and Age Factors in High Fat Diet-Induced Obesity in C57BL/6J and BALB/cA Mice. Exp. Anim. 2007, 56, 263–272, doi:10.1538/expanim.56.263.
  4. Jayaprakasam, B.; Olson, L.K.; Schutzki, R.E.; Tai, M.-H.; Nair, M.G. Amelioration of Obesity and Glucose Intolerance in High-Fat-Fed C57BL/6 Mice by Anthocyanins and Ursolic Acid in Cornelian Cherry ( Cornus Mas ). J. Agric. Food Chem. 2006, 54, 243–248, doi:10.1021/jf0520342.
  5. Li, X.; Yu, X.; Sun, D.; Li, J.; Wang, Y.; Cao, P.; Liu, Y. Effects of Polar Compounds Generated from the Deep-Frying Process of Palm Oil on Lipid Metabolism and Glucose Tolerance in Kunming Mice. J. Agric. Food Chem. 2017, 65, 208–215, doi:10.1021/acs.jafc.6b04565.
  6. Norris, G.H.; Porter, C.M.; Jiang, C.; Millar, C.L.; Blesso, C.N. Dietary Sphingomyelin Attenuates Hepatic Steatosis and Adipose Tissue Inflammation in High-Fat-Diet-Induced Obese Mice. The Journal of Nutritional Biochemistry 2017, 40, 36–43, doi:10.1016/j.jnutbio.2016.09.017.

Doxycycline showed pharmacological characteristics that would allow its incorporation as adjuvant treatment in the prevention of diabetes and atherosclerosis in preclinical stages. An adverse effect of antibiotics on the host refers to damage to the microbiota, generating alterations in the metabolism of sugars and lipids. In this study, this factor was not controlled [49]. The gut microbiome plays an important role in extracting energy from food and inducing obesity [50]. It is conceivable that the effect of doxycycline on the microbiota could interfere with the body weight behavior of the animal model. Furthermore, it is important to note that a limitation of the non-transgenic animal model used is that it does not develop a significant increase in body weight from a chronically high-fat diet. Therefore, the effect of doxycycline on body weight in this model may not be adequate. It would be desirable to carry out future studies with other animal models and preserving the intestinal microbiota. One of the strategies that could be established is the structural modification of doxycycline in its “A ring” (antimicrobial activity), reducing its action on the microbiota without losing its effects on other molecular targets [51].

  1. Ahn, Y.; Jung, J.Y.; Kweon, O.; Veach, B.T.; Khare, S.; Gokulan, K.; Piñeiro, S.A.; Cerniglia, C.E. Impact of Chronic Tetracycline Exposure on Human Intestinal Microbiota in a Continuous Flow Bioreactor Model. Antibiotics 2021, 10, 886, doi:10.3390/antibiotics10080886.
  2. Davis, C.D. The Gut Microbiome and Its Role in Obesity. Nutr Today 2016, 51, 167–174, doi:10.1097/NT.0000000000000167.
  3. Fuoco, D. Classification Framework and Chemical Biology of Tetracycline-Structure-Based Drugs. Antibiotics 2012, 1, 1–13, doi:10.3390/antibiotics1010001.

Comment: Additionally, for Comment 3, the response of "data not shown" is unacceptable and it would be better to provide adequate evidence to support the conclusion.

Answer: Thank you for your detailed and accurate comment. The data are now showed in table 1.

Reviewer 2 Report

The current revised version presents significant improvement and more critically reviewed aspects on the introduction, presenting the model design, interpretation of data, limitations, and future directions. All together will make the manuscript more accurate, critically presented, with better fit in the "big picture" and inviting/attracting minds/researchers to catch and come up with rationale scientific questions in several aspects of this topic to verify and pursue.

However, there are still one review that needs to be added and some typographical errors to be edited.

1.In Discussion: Line 417, right after: “doxycycline administration did not reduce body weight”. It is more informative to come up with/think of adding few lines to explain/present possibilities behind the “no change in body weight among the groups”. I suggest this can be due to either or both of two possibilities. First: gut microbiome plays a major role in energy extraction from food and obesity induction. It is conceivable that the microbiome was partially but not totally consistent or controlled among the groups (see Cindy D. Davis review: The gut microbiome and its role in obesity Nutrition Today July-Aug;51(4):167-174, and others). Second: In this present study, the experimental settings including mouse strain, subtle difference in elements or fibers in the SD and HFD diets all together did not affect inducing atherosclerosis and hyperglycemic pathways but interfered with induction and/or response to doxycycline in the pathways of adipogenesis, lipidemia and  obesity pattern of weight gain. In such scenario, Doxycycline’s effect was consequently revealed only on atherosclerosis.  

 This above-mentioned paragraph can be certainly integrated (incorporated) with the present explanation in this V1 of the manuscript in line 416-420.    

 1. Line 74: modify to: was not possible to establish (remove "ed") 

2. Line 379: Furthermore, to its antibacterial properties doxycycline also exhibits anti-inflammatory.

comment: edit to: Further to its antibacterial properties, doxycycline also exhibits…. remove “more.” OR use: In addition to its antibacterial properties, doxycycline…

 3. Line 204, 205: Animals were kept fasting overnight and glucose was injected IP

 4.Line 239: correct to be: healthy mice (SD), fed with high fat diet for six months…. (remove whose). Alternatively, you can draft as: healthy mice (SD) that were fed with high fat diet…

 5. In line 257: pasting from the text: “ With the above, it can be said that doxycycline is not able. Consider redrafting as: Given the above, one can conceive that doxycycline is not able to…..OR we can conceivably state that doxycycline was not able to....

 6. Line 259: change it could  prevent to: but it prevented preclinical atherosclerosis OR but it apparently prevented preclinical atherosclerosis.

 7. It is appreciated β-cells to avoid possible confusing of ‘immunology-oriented readers with B-cells, commonly used for B-lymphocytes. Yet, in final editing consistently replace all other B-cells with β-cell or beta-cells in the whole text and tables.

Author Response

Comment: The current revised version presents significant improvement and more critically reviewed aspects on the introduction, presenting the model design, interpretation of data, limitations, and future directions. All together will make the manuscript more accurate, critically presented, with better fit in the "big picture" and inviting/attracting minds/researchers to catch and come up with rationale scientific questions in several aspects of this topic to verify and pursue.

However, there are still one review that needs to be added and some typographical errors to be edited.

In Discussion: Line 417, right after: “doxycycline administration did not reduce body weight”. It is more informative to come up with/think of adding few lines to explain/present possibilities behind the “no change in body weight among the groups”. I suggest this can be due to either or both of two possibilities. First: gut microbiome plays a major role in energy extraction from food and obesity induction. It is conceivable that the microbiome was partially but not totally consistent or controlled among the groups (see Cindy D. Davis review: The gut microbiome and its role in obesity Nutrition Today July-Aug;51(4):167-174, and others). Second: In this present study, the experimental settings including mouse strain, subtle difference in elements or fibers in the SD and HFD diets all together did not affect inducing atherosclerosis and hyperglycemic pathways but interfered with induction and/or response to doxycycline in the pathways of adipogenesis, lipidemia and  obesity pattern of weight gain. In such scenario, Doxycycline’s effect was consequently revealed only on atherosclerosis.  

 This above-mentioned paragraph can be certainly integrated (incorporated) with the present explanation in this V1 of the manuscript in line 416-420.    

Answer: Thank you for your detailed and accurate comment. All of your observations were incorporated into the discussion section. A summary of what you observe is analyzed and recommendations are made for future investigations based on what you observed. The suggested paragraph was incorporated as follows:

Doxycycline showed pharmacological characteristics that would allow its incorporation as adjuvant treatment in the prevention of diabetes and atherosclerosis in preclinical stages. An adverse effect of antibiotics on the host refers to damage to the microbiota, generating alterations in the metabolism of sugars and lipids. In this study, this factor was not controlled [49]. The gut microbiome plays an important role in extracting energy from food and inducing obesity [50]. It is conceivable that the effect of doxycycline on the microbiota could interfere with the body weight behavior of the animal model. Furthermore, it is important to note that a limitation of the non-transgenic animal model used is that it does not develop a significant increase in body weight from a chronically high-fat diet. Therefore, the effect of doxycycline on body weight in this model may not be adequate. It would be desirable to carry out future studies with other animal models and preserving the intestinal microbiota. One of the strategies that could be established is the structural modification of doxycycline in its “A ring” (antimicrobial activity), reducing its action on the microbiota without losing its effects on other molecular targets [51].

  1. Ahn, Y.; Jung, J.Y.; Kweon, O.; Veach, B.T.; Khare, S.; Gokulan, K.; Piñeiro, S.A.; Cerniglia, C.E. Impact of Chronic Tetracycline Exposure on Human Intestinal Microbiota in a Continuous Flow Bioreactor Model. Antibiotics 2021, 10, 886, doi:10.3390/antibiotics10080886.
  2. Davis, C.D. The Gut Microbiome and Its Role in Obesity. Nutr Today 2016, 51, 167–174, doi:10.1097/NT.0000000000000167.
  3. Fuoco, D. Classification Framework and Chemical Biology of Tetracycline-Structure-Based Drugs. Antibiotics 2012, 1, 1–13, doi:10.3390/antibiotics1010001.

Comment: 1. Line 74: modify to: was not possible to establish (remove "ed") 

Answer: Thanks for the observation, the sentence was rewritten.

Comment: 2. Line 379: Furthermore, to its antibacterial properties doxycycline also exhibits anti-inflammatory. Comment: edit to: Further to its antibacterial properties, doxycycline also exhibits…. remove “more.” OR use: In addition to its antibacterial properties, doxycycline…

Answer: Thanks for the observation, the sentence was rewritten.

Comment: 3. Line 204, 205: Animals were kept fasting overnight and glucose was injected IP

Answer: Thanks for the observation, the sentence was rewritten.

Comment: 4. Line 239: correct to be: healthy mice (SD), fed with high fat diet for six months…. (remove whose). Alternatively, you can draft as: healthy mice (SD) that were fed with high fat diet…

Answer: Thanks for the observation, the sentence was rewritten.

Comment:  5. In line 257: pasting from the text: “ With the above, it can be said that doxycycline is not able. Consider redrafting as: Given the above, one can conceive that doxycycline is not able to…..OR we can conceivably state that doxycycline was not able to....

Answer: Thanks for the observation, the sentence was rewritten.

Comment: 6. Line 259: change it could prevent to: but it prevented preclinical atherosclerosis OR but it apparently prevented preclinical atherosclerosis.

Answer: Thanks for the observation, the sentence was rewritten.

Comment: 7. It is appreciated β-cells to avoid possible confusing of ‘immunology-oriented readers with B-cells, commonly used for B-lymphocytes. Yet, in final editing consistently replace all other B-cells with β-cell or beta-cells in the whole text and tables.

Answer: Thanks for the observation, Thanks for the comment, suggested changes were made to the text to appropriately refer to beta cells.

Reviewer 4 Report

The paper has much improved. It can be further processed for publication. Please watch accurately the tables after deleting the track version as something could be missed.

Author Response

Comment: The paper has much improved. It can be further processed for publication. Please watch accurately the tables after deleting the track version as something could be missed.

Answer: Thanks for the observation and suggestions. The manuscript was carefully revised after acceptance of the changes to avoid loss of values in tables and text.

Round 3

Reviewer 1 Report

No more comments